# Dietary Choline Mitigates High-Fat Diet-Impaired Chylomicrons Assembly via UPRer Modulated by *perk* DNA Methylation

**DOI:** 10.3390/cells11233848

**Published:** 2022-11-30

**Authors:** Zhen-Yu Bai, Hua Zheng, Zhi Luo, Christer Hogstrand, Ling-Jiao Wang, Yu-Feng Song

**Affiliations:** 1Key Laboratory of Freshwater Animal Breeding, Ministry of Agriculture, Fishery College, Huazhong Agricultural University, Wuhan 430070, China; 2Laboratory for Marine Fisheries Science and Food Production Processes, Qingdao National Laboratory for Marine Science and Technology, Qingdao 266237, China; 3Department of Nutritional Sciences, School of Medicine, King’s College London, Franklin-Wilkins Building, 150 Stamford Street, London SE1 9NH, UK

**Keywords:** high-fat diet, choline, chylomicrons, perk, Apob48, methylation

## Abstract

High-fat diets (HFD) lead to impairment of chylomicrons (CMs) assembly and adversely influence intestinal lipid homeostasis. However, the mechanisms of HFD impairing CMs assembly have yet to be fully understood. Additionally, although choline, as a lipid-lowering agent, has been widely used and its deficiency has been closely linked to non-alcoholic steatohepatitis (NASH), the contribution of choline by functioning as a methyl donor in alleviating HFD-induced intestinal lipid deposition is unknown. Thus, this study was conducted to determine the mechanism of HFD impairing CMs assembly and also tested the effect of choline acting as a methyl donor in this process. To this end, in this study, four diets (control, HFD, choline and HFD + choline diet) were fed to yellow catfish for 10 weeks in vivo and their intestinal epithelial cells were isolated and incubated for 36 h in fatty acids (FA) with or without choline solution combining si-*perk* transfection in vitro. The key findings from this study as follows: (1) HFD caused impairment of CMs assembly main by unfolded protein response (UPRer). HFD activated *perk* and then induced UPRer, which led to endoplasmic reticulum dysfunction and further impaired CMs assembly via protein–protein interactions between Perk and Apob48. (2) Choline inhibited the transcriptional expression level of *perk* via activating the −*211* CpG methylation site, which initiated the subsequent ameliorating effect on HFD-impaired CMs assembly and intestinal lipid dysfunction. These results provide a new insight into direct crosstalk between UPRer and CMs assembly, and also emphasize the critical contribution of choline acting as a methyl donor and shed new light on choline-deficient diet-induced NASH.

## 1. Introduction

The continued epidemic of obesity has seen focused attention being paid to the metabolic pathways and key players that regulate dietary lipid absorption from the small intestine [1,2]. Chylomicrons (CMs) are specialized vehicles synthesized by the intestine to transport large quantities of dietary fat and fat-soluble vitamins [1,3,4]. Thus, CMs synthesis and circulation are essential for adequate absorption of lipid in all tissues, and are also the basis for ensuring intestinal lipid metabolic homeostasis [5], whereas CMs assembly and secretion are vulnerable to a wide range of nutritional factors [6]. CMs’ size has been found to be significantly dependent on diet conditions, especially the amount of fat absorbed from a diet [7]. Thus, increasing evidence shows that a high-fat diet (HFD), an increasingly common diet worldwide with potential obesity risk factors, has been closely associated with the impaired packaging and export of CMs [7,8]. However, the molecular mechanisms governing HFD blocking CMs assembly and secretion have yet to be fully understood.

The process of CMs assembly and secretion is a multifaceted and complex process. The intestinal assembly of CMs is initiated by the lipidation of apolipoprotein B (APOB) by microsomal TG transfer protein (MTP) in the endoplasmic reticulum (ER) of the enterocytes to form pre-chylomicron particles [4,6]. Thus, MTP as an essential protein to promote CMs assembly by shuttling TG, CE, and PL to the structural ApoB-48 [9,10,11]. In addition, secretion-associated Ras-related GTPase 1B (Sar1B) is another crucial component of COPII vesicles that starts from the ER to transport pre-CMs to the Golgi apparatus [12]. On the other hand, since ER is the initial and main site for CMs assembly, ER dysfunction has been closely connected to dysregulation of the CMs biogenesis [13,14,15]. Naturally, as a conserved adaptive mechanism in response to ER dysfunction, ER-unfolded protein response (UPRer) has been found to be involved in the regulation of CMs assembly [13,16], although the relevant mechanism remains unresolved. Meanwhile, multiple investigators suggested HFD is the most common inducement to UPRer [17,18], as seen in our previous study in fish [19]. However, few studies regarding the mechanism behind HFD-induced UPRer impairing CMs assembly and secretion have been reported.

Nowadays, non-alcoholic steatohepatitis (NASH) is one of the fastest growing liver diseases, characterized by hepatic lipid dysregulation [20,21]. Choline as an essential dietary nutrient plays a vital role in maintaining hepatic lipid homeostasis, thus its deficiency has been closely linked to NASH [22,23,24]. On the other hand, although the lipid-lowering effects of dietary choline for HFD-induced lipid accumulation have been widely studied and also experimentally confirmed [25,26,27], previous studies neglected the contribution of choline acting as a methyl donor in this process. Choline serves as a methyl donor by its role as a precursor of acetylcholine, which can directly modulate DNA methylation [28,29]. DNA methylation represents one of the mechanisms that contributes to dysregulation of gene expression via interaction with nutritional factors [28,30]. Furthermore, the close association between genome-wide DNA methylation and HFD-induced lipid metabolic disorder has been investigated [31,32]. Thus, it is reasonable to speculate on the potential regulatory role of choline-mediated DNA methylation for HFD-induced lipid dysregulation. On the other hand, studies indicated that choline improved intestinal lipid metabolism via regulating CMs lipoprotein secretion [33,34,35], implying the directly causal relationships between CMs assembly and intestinal lipid homeostasis. Moreover, the involvement of UPRer-related DNA methylation in HFD causing abnormal lipid metabolism has also been investigated. These previous studies suggest that choline ameliorated HFD-induced intestinal lipid metabolic disorder might by mediating CMs assembly via UPRer-related DNA methylation, although the underlying mechanism is unclear.

The yellow catfish *Pelteobagrus fulvidraco* is an economically important freshwater teleost fish and is widely distributed in China and other countries [36]. This species is commonly used as a good model for studying lipid metabolism because of the similarities of its metabolic and regulatory pathways to those of mammals and the high degree of similarity of its genome to that of humans [36,37]. Moreover, our previous studies found that HFD caused severe UPRer and subsequently induced intestinal lipid metabolic disorder [24]. Here, given the central role of CMs biogenesis for intestinal lipid metabolism, it is meaningful to explore the effects and mechanisms of HFD-induced UPRer in CMs assembly and its role in intestinal lipid metabolic dysfunction. Additionally, although the lipid-reducing capacity of choline has been revealed in our previous studies [26], the potential role of choline in HFD-mediated UPRer and intestinal lipid dysregulation when it functions as a methyl donor still remains unknown. Thus, the purpose of this study was to test the mechanism of HFD-induced UPRer-impairing CMs biogenesis, and also to uncover the role of choline acting as a methyl donor in this process. Our present study reveals the important regulatory role of Perk, as critical sensors and transducers for UPRer signaling pathways in HFD impairing CMs assembly through protein–protein interaction with Apo48. Meanwhile, the novel alleviating function of choline in this process via the site-specific DNA methylation of *perk* was also found. Our study will provide a better understanding of the molecular mechanism of CMs assembly and secretion and their potential role in HFD-induced lipid metabolic dysfunction, which also shed new light on choline-deficient diet-induced NASH.

## 2. Materials and Methods

### 2.1. Ethical Statement

Huazhong Agricultural University’s (HZAU) institutional ethical guidelines for the care and use of laboratory animals were followed throughout all investigations, and were approved by the Ethical Committee of HZAU (identification code: Fish-2020-07-21).

### 2.2. Expt. 1: Animals Feeding, Management, and Sampling

Dietary formula and yellow catfish feeding were determined according to our previous studies [19,26]. We formulated four experimental diets, as shown in Appendix A. Dietary lipid concentrations were 10.3% (control group), 14.5% (high-fat diet, HFD) group, 10.3% (choline diet group, CH), and 14.3% (high-fat diet + choline, HFD + CH), respectively. Fish oil and soybean oil (1:1, *w/w*) were used as the lipid sources. The addition of choline chloride was 563.4 (Control), 578.9 (HFD), 1650.1 (CH), and 1652.3 (HFD + CH) mg of choline per kg diet (≥99.0% in purity, Sinopharm Chemical Reagent Co., Ltd., Shanghai, China). All the fish were obtained from a local fish farm (Wuhan, China) and transferred into 300-litre circular tanks for 2 weeks to acclimatize. At the start of the study, each tank was supplied with thirty fish of uniform size (average starting weight: 3.83 ± 0.13 g, mean ± SEM). Each food item was randomly assigned to the three water tanks, and 12 tanks were used in the experiment. Daily mortality statistics were kept. The experiment lasted for ten weeks. Water temperature, dissolved oxygen (DO), pH, and ammonia nitrogen (NH4-N) were measured twice weekly and their values were 28.5 ± 0.41 °C, 6.42 ± 0.02 mg/L, 7.65 ± 0.17, and 0.0812 ± 0.06 mg/L, respectively. The morphological parameters and growth performance were shown in Appendix A, which also can be checked in our parallel in our parallel study [38].

To avoid prandial effects, all yellow catfish were fasted for 24 h after ten weeks of feeding. MS-222 was used to euthanize all of the yellow catfish. All yellow catfish were counted and weighed in bulk. The intestines of three fish from each tank were sampled for histological and ultrastructural examination, respectively. Other fish intestinal samples were promptly frozen in liquid nitrogen for further examination, including the contents of TG, MTP activity, density gradient analysis and chylomicron extraction, methylation analysis, and gene and protein expression. After resting for 4 h, blood samples were drawn from the caudal vein and centrifuged at 4000× *g* and 4 °C for 10 min to produce serum samples. The intestines were immediately frozen in liquid nitrogen and kept at −80 °C until further analysis.

### 2.3. Expt. 2: Cell Culture and Treatments

Primary intestinal epithelial cells (IECs) were isolated from juvenile yellow catfish, and cultured according to the protocols in our earlier publication [19]. To explore the impact of HFD on intestinal CMs assembly and lipid metabolism in vitro, the IECs were treated with palmitic acid (PA) and oleic acid (OA) at a ratio of 1:1. The results of the MTT experiment demonstrated that the viability of IECs was unaffected by FA and CH at concentrations of up to 0.5 mM and 70 M, respectively. FA incubation (0.4 mM) increased the TG content, and CH incubation (60 μM) decreased the TG content (Appendix A). Thus, 0.4 Mm FA and 60 μM CH concentration were used in the in vitro experiment. In brief, four experimental treatments were devised: the control group (no additional addition), the FA group (0.4 mM), the CH group (60 mM), and the FA + CH group (0.4 mM of FA + 60 mM of choline). The cells were incubated at 28 °C for 36 h. Three fish were utilized to create a pool of cells for each cell culture. To explore the effects and mechanisms of Perk in CMs assembly, a small interfering RNA (siRNA) was designed specifically for the *perk* gene in IECs. Furthermore, in order to identify the contribution of dietary choline acting as a methyl donor in alleviating HFD-induced impairment of CMs assembly, we used the pharmacological methylation inhibitor 5-azacytidine (HY-10586; MedChemExpress, Monmouth Junction, NJ, USA) to incubate the IECs.

Human embryonic kidney cells (HEK293T cells) have high transfection efficiency and have been widely used to explore the genetic function in fish [39]. In the present study, HEK293T cells were incubated into the control (without extra addition), FA (0.4 mM), CH (60 μM), and FA + choline (0.4 mM of FA + 60 μM of choline) for 36 h to explore the effect of special *perk* methylation sites in CMs assembly.

### 2.4. Sample Analysis

#### 2.4.1. ORO, H&E, Bodipy 493/503 Staining and Transmission Electron Microscopy (TEM) Observation

Oil red O (ORO) and Hematoxylin and Eosin (H&E) staining tests were conducted according to the manufacture’s instruction, and the analytical protocols followed the description in our previous publications [39,40]. The software Image J (NIH, Bethesda, MD, USA) was used to quantify the relative areas of lipid droplets (LDs) in the ORO staining, as well as the relative intestinal villus length and muscular layer thickness in the H&E staining.

Bodipy 493/503 staining for LDs and TEM observation of IECs were carried out according to the protocol described in our previous publications [19,39]. IECs were cultured in 12-well plates for 36 h, then washed twice with PBS and incubated with the 5 g mL^−1^ Bodipy 493/503 (D3922; Thermo Fisher Scientific, Waltham, MA, USA) for 30 min, followed by three PBS washes. A laser scanning confocal microscope (Leica DMI8) was used to visualize the fluorescence intensity in the IECs. The green dots represent LDs. FlowJo v.10 software was used for data analysis.

#### 2.4.2. Cell Viability, TG Contents, MTP Activity

Cell viability was assessed using 3-(4,5-dimethylthiazol-2-yl)-2, 5-diphenyltetrazolium bromide (V13154; Thermo Fisher Scientific) in our previous publication [19]. According to the manufacturer’s instructions, commercial kits (A110-1-1, Nanjing Jiancheng Bi-oengineering Institute, Nanjing, Jiangsu, China) were used to determine the contents of TG. Bradford protocols [41] were used to determine the soluble protein content. MTP activity was determined using a commercially available fluorescence assay kit (MAK110-1KT, Roar Biomedical Inc., New York, NY, USA). These tests were carried out in triplicate.

#### 2.4.3. RNA Isolation and Quantitative Real-Time PCR Analysis (qPCR)

A Trizol reagent (15596026, Thermo Fisher Science) was used to isolate total RNA, which was then reverse-transcribed into cDNA using the Reverse Transcription Kit (6110A, TaKaRa, Tokyo, Japan,). qPCR assays were carried out in accordance with our published protocol [19]. *Gapdh, b2m, ef1α, 18s rRNA, tuba, β-actin, hprt1, ubc9,* and *tbp* were utilized as the nine housekeeping genes to examine the stability of their mRNA expression. According to the analysis performed by geNorm (https://genorm.cmgg.be/, accessed on 20 May 2021), the geometric mean of the best two gene combinations was used to normalize the relative expression of each gene, which was obtained using the 2^−ΔΔCt^ approach, as in our prior studies [39].

#### 2.4.4. Western Blot

According to our previous research [39,40], we employed Western blot to determine protein levels. In brief, the intestines and cells were lysed in a RIPA buffer (89900, Thermo Fisher Scientific) with a PMSF protease inhibitor (36978, Thermo Fisher Scientific). SDS-PAGE gel was used to load the protein, which was subsequently transferred to the PVDF membranes. The membranes were blocked with skimmed milk (8%, *w*/*v*), and then incubated with one of the primary antibodies listed below overnight at 4 °C, including anti-Gapdh (1:10,000, 12,118, Cell Signaling Technology, Danvers, MA, USA), anti-Mtp (1:1000, 14,028, Cell Signaling Technology), anti-Dgat1 (1:1000, A6857, ABclonal, Wuhan, Hubei China), anti-Perk (1:1000, 8196, Cell Signaling Technology;), anti-Dnmt1 (1:1000, 5032, Cell Signaling Technology), and anti-Apob (1:1000, A1330, ABclonal). The secondary antibodies, including an HRP-conjugated anti-rabbit IgG antibody (1:10,000, 7074, Cell Signaling Technology), were then incubated with the membranes. The membranes were seen by ECL (1705060, Bio-Rad; Hercules, CA, USA) after additional washing. The membranes were seen using enhanced chemiluminescence, and Image J was used to measure the densitometry of these bands.

#### 2.4.5. Bisulfite Sequencing of Perk

To determine the methylation level of *perk*, the *perk* gene was sequenced by bisulfite based on our published protocol [42]. In brief, a tissue DNA kit (D3396-00S, Omega Biotek, Norcross, GA, USA) was used to separate genomic DNA from yellow catfish intestinal samples, and then the DNA Methylation-Gold Kit (D5006, Omega Biotek, Norcross, GA, USA) was used to modify them according to the manufacturer’s recommendations. The bisulfite-modified DNA was then amplified using PCR using particular primer pairs. The operation of PCR included 95 °C denaturation for 5 min, and then 40 cycles of 95 °C for 30 s, 56 °C for 30 s, and 72 °C for 35 s, and at the end of 72 °C extension for 5 min. The purified PCR products were then cloned into pMD19-T vectors (6013, TaKaRa). Nine clones from each sample were chosen after cloning and utilized for DNA sequencing. The sequencing analysis was done by TsingKe Bio-logical Technology (Wuhan, China).

#### 2.4.6. Chylomicron Isolation

The protocol of chylomicron isolation was taken from previous research and modified [43]. Briefly, intestinal tissue 0.1~0.2 g was taken and mixed into a homogenate with PBS buffer solution. The precipitates were resuspended by centrifugation several times, and the supernatants were collected. Secondly, the adjusted supernatant density was 1.10 g·mL^−1^, successively covered with 3.75 mL 1.063 g·mL^−1^, 3.75 mL 1.019 g·mL^−1^, 2.5 mL 1.006 g·mL^−1^ NaCl density gradient medium, 15 °C, 345,000 g min^−1^, 1.5 h, carefully absorbing 2.5 mL volume of liquid from the upper layer, which are CMs.

#### 2.4.7. Immunofluorescence Analysis

Immunofluorescence was utilized to analyze the colocalization of Apob and LDs in the IECs, according to our most recent publications [40,42]. In brief, cells were cultivated in 12 well-plates and treated for 36 h before being rinsed three times with PBS and preserved in 4% paraformaldehyde at room temperature for 10 min, and then centrifuged at 850× *g* min^−1^ for 10 min to collect the cells. Following this, they were incubated with particular primary antibodies, such as rabbit anti-Apob (1:200), and overnight at 4 °C after being blocked in 5% BSA for 2 h. Following three 5-min PBST washes, the cells were incubated for 60 min at room temperature and in the dark with a secondary anti-body made of goat anti-rabbit IgG H&L (AlexaFluor^®^ 647, 1:500, 150079; Abcam, MA, USA). The pictures were captured using a laser scanning confocal microscope (Leica Wetzlar, Germany). The CytoFlex flow cytometer (Beckman Coulter, Miami, FL, USA) was used to measure the LDs, which were identified as the green dots. The software FlowJo v.10 was used to analyze the data.

#### 2.4.8. Small Interference RNA Transfection

Regarding small interfering RNA suppression of *perk* (siRNA) according to our most recent study [33], *perk* genes were knocked down in yellow catfish IECs using particular short interfering RNA. In other words, IECs were extracted and grown as previously mentioned. The EntransterTM-R4000 Reagent kit (4000-4, Engreen Biosystem; Portland, USA) was used to transfect the appropriate siRNAs in accordance with the manufacturer’s instructions. In a 12-well culture plate, 50 nM siRNA duplex (Gene Pharma, Shanghai, China) and 2 μL of the Entranster TM-R4000 Reagent kit were diluted in Opti-MEMTM (31985088, Thermo Fisher Scientific). The final step was adding the mixture to the cell culture for 36 h. RT-qPCR and Western blot were used to determine the knockdown effectiveness of *perk*, and the siRNA with the highest knockdown efficiency was used in our research. (Appendix A, Appendix A). The control siRNA was a non-silencing siRNA. Gene Pharma provided all the siRNA sequences for *perk*.

#### 2.4.9. Plasmid Construction, Cloning of Promoters, Transfections, and Luciferase Assays

First, we cloned the sequences of perk promoters disclosed by Xu et al. [42], based on the published draft genome of yellow catfish. Next, we created *perk* promoters using the ClonEx-pressTM II One Step Cloning Kit and inserted them into the pGl3 basic vector (C112, Vazyme, Piscataway, NJ, USA). The CpG island of *perk* was predicted by the MethPrimer (http://www.urogene.org/methprimer/, accessed on 25 May 2021, Appendix A). Appendix A presents the promoter cloning primers that were employed. According to the manufacturer’s instructions for the Mut Express II Fast Mutagenesis Kit (C214-01, Vazyme), mutations of CG sites in the promoter regions were carried out and validated by sequencing. To assess the luciferase activity, all of these plasmids were transiently transfected into HEK293T cells using Lipofectamine 2000 (12566014, Invitrogen; Carlsbad, CA, USA). Primers used for site-mutation analysis are presented in Appendix A.

#### 2.4.10. Immunoprecipitation

According to our prior publications [40,44], immunoprecipitation (IP) tests were carried out to examine the interaction between Perk and Apob48. For IP analysis, yellow catfish IECs were lysed in NP-40 buffer (p0013F, Beyotime Institute of Biotechnology; Shanghai, China) supplemented with a protease inhibitor cocktail (P1010, Beyotime Institute of Biotechnology), and then the cell lysate was incubated overnight at 4 °C with anti-Perk (1:1000, ab59256, Abcam), anti-Apob (1:1000, ab18181, Abcam), followed by addition of the protein A/G beads (P2012, Beyotime Institute of Biotechnology). Immunocomplexes were washed 5 times with NP-40 buffer and used for WB analysis.

#### 2.4.11. The Prediction of Perk-Apob48 Structural Protein Models

The three-dimensional structures of the target proteins were obtained from the Protein Data Bank (http://www.rcsb.org/pdb/home/home.do, accessed on 15 May 2021) to obtain the Perk-Apob48 crystal structure with ligand and high resolution. The crystal structure with ligand and high resolution was obtained. The Clean Protein tool in Discovery Studio 4.0 (DS) was used to perform the following operations on the protein structure: deletion of ligand and water molecules, completion of incomplete residues, removal of redundant protein conformation, and addition of hydrogen atoms and Gasteiger–Marsili charges.

### 2.5. Statistical Analysis

Statistical analysis was performed with the SPSS 19.0 (IBM, Armonk, NY, USA). At first, results were expressed as mean ± S.E.M (n ≥ 3). Before statistical analysis, all dates were evaluated for normality with the Kolmogorov–Smirnov test, and the Bartlett’s test was conducted to test the homogeneity of the variances among the treatments. 2-factor ANOVA was applied to assess the effects of HFD, CH, and their interactions in in vivo studies. When a 2-factor ANOVA revealed a significant P-interaction (*p* < 0.05), the differences between the groups were assessed with Duncan’s multiple range test. The one-factor ANOVA with Duncan’s multiple range test was used to analyze the data of the in vitro study among the same si-NC or si-*perk* groups/−AZA or +AZA groups. Different lower-case letters indicated significant differences in si-NC/−AZA groups, and different capital letters indicated significant differences in si-perk/+AZA groups (*p* ≤ 0.05). Student’s *t*-tests were undertaken to analyze the data of the in vitro study between si-NC and si-perk groups/−AZA and +AZA groups.

## 3. Results

### 3.1. Dietary Choline Alleviated HFD-Induced Intestinal Lipid Dysregulation

In the present study, HFD caused intestinal structural damage supported by a shorter intestinal villi height and a thicker muscular layer thickness when compared to the control diet group (Figure 1A–C). Meanwhile, HFD increased intestinal LDs amounts and also induced excessive TG accumulation (Figure 1A,D,E). Further studies indicated HFD significantly up-regulated the expression of genes involved in FA re-esterification (*dgat1, dgat2, mgat1, mgat2*), but not de novo FA synthesis (*fas, g6pt, 6gpt*) (Figure 1F). This suggested that FA re-esterification is the main driving force behind HFD-induced intestinal TG accumulation, which was further proven by the up-regulated Dgat1 protein expression in the HFD group (Figure 1G). On the other hand, dietary choline markedly alleviated HFD-induced intestinal structural damage and lipid dysregulation also via the main FA re-esterification.

### 3.2. Alleviated Effects of Dietary Choline on HFD-Induced Impairment of CMs Assembly

Given that FA re-esterification is the principal pathway for CMs-TG production, we next investigated the effects of HFD on CMs assembly and the role of dietary choline in this process. First, compared to control diet group, HFD significantly decreased the size and quantity of CMs in TEM observations, and also caused the decline of CMs lipoprotein and TG content in both the intestines and serum (Figure 2A–G and Appendix A). Second, HFD significantly down-regulated the expression level of key genes and protein involved in CMs assembly, including *apob, apobec1, mtp, sar1b, cd36,* and *vamp7* at the transcriptional level (Figure 2H), and Apob48, Mtp, and Sar1b protein expression level (Figure 2I). In addition, the notable down-regulation of MTP activity was found in the HFD group (Figure 2J). All these suggested that HFD could induce impairment of CMs assembly in the intestine of yellow catfish. Additionally, dietary choline markedly attenuated HFD-induced impairment of CMs assembly, as evidenced by the increased CMs-lipoprotein/TG content, and also the up-regulated expression levels of CMs assembly-related key genes and proteins.

### 3.3. The Involvement of Perk Signaling-Mediated UPRer in Choline Attenuating HFD-Induced Impairment of CMs Assembly

Since CMs assembly begins in the rough ER, CMs assembly depends on the normal ER function. Thus, next we examined whether UPRer as a conserved mechanism in response to ER dysfunction participated in HFD/choline-mediated CMs assembly. Compared to the control group, HFD caused severe ER swelling and also significantly upregulated the mRNA levels of UPRer-associated marker genes, including *grp78, perk, eif2a, ire1a, xbp1,* and *atf6* (Figure 3A,B). It is noteworthy that among these UPRer-associated markers, *perk* and its downstream factor *eif2a* were the most sensitive to HFD, which was further confirmed by the protein expression of Perk and Grp78 (Figure 3C). These results indicated HFD caused the main UPRer via *perk* signaling in the intestines of yellow catfish. Meanwhile, the *perk*-signaling-mediated UPRer was also markedly prevented by the addition of dietary choline.

Next, we examined the role of *perk*-signaling-mediated UPRer on CMs assembly using isolated primary intestinal epithelial cells (IECs) from yellow catfish with si-*perk* transfection. As shown in Figure 4A–G, si-*perk* significantly improved FA-induced impairment of CMs assembly evidenced by the enhancement of the size and quantity of CMs, CMs-lipoprotein, and TG content in IECs. Meanwhile, si-*perk* transfection also decreased FA-induced down-regulation of Apob48 (Figure 4H,I). In addition, *perk* inhibition caused further improvement of choline-accelerated CMs assembly (Figure 4H,I). These results suggested the regulatory role of *perk* on FA-induced impairment of CMs assembly and choline-activated ameliorating effects. On the other hand, the activation of CMs assembly induced by si-*perk* was accompanied by mitigation of LDs accumulation under FA/choline incubation (Figure 4J). This indicated the directly causal relationships between CMs assembly and intestinal lipid homeostasis in the IECs of yellow catfish.

### 3.4. Perk-Apob48 Interaction Is Required for Perk-Signaling-Mediating CMs Assembly

ApoB48, a major chylomicron apoprotein, is essential to CMs assembly. Thus, to further investigate the regulatory mechanism of *perk*-signaling-mediating CMs assembly, we next focus on the interrelation between Perk and Apob48 and its roles on CMs assembly. First, *perk*-knockdown significantly up-regulated the expression of CMs assembly-associated markers, including *apob*, *mtp,* and *sar1b* at transcriptional levels, and also Apob48, Mtp, and Sar1b at protein levels and MTP activity (Figure 5A–D). This further confirmed the upstream regulatory role of *perk* signaling in CMs assembly. Second, protein–protein binding sites prediction demonstrated that Perk had coupling effect domain with Apob48 (Figure 5E), and further IP analysis showed that Perk was co-precipitated with Apob (Figure 5F). This provided the first experimental evidence for direct crosstalk between perk signaling and CMs assembly. Importantly, this Perk-Apob48 interaction was apparently up-regulated by FA incubation, but down-regulated by the addition of choline (Figure 5F), suggesting the involvement of Perk-Apob48 interaction in choline-ameliorating FA-induced impairment of CMs assembly.

### 3.5. Choline Down-Regulated Perk Expression Controlled by −211 Site-Specific DNA Methylation

After we determined the ameliorating role of choline on HFD/FA-induced impairment of CMs assembly and intestinal lipid dysregulation, we next investigated the contribution of choline acting as a methyl donor in *perk*-mediated CMs assembly. First, in vivo dietary choline significantly relieved the inhibitory effects of HFD on mRNA and/or protein levels of DNA methyltransferase (*dnmt1, dnmt3a, and dnmt3b*) (Figure 6A,B), and also markedly increased the whole DNA methylation level of *perk* (Figure 6C and Appendix A). Next, we ascertained the specific methylation site of *perk* mediated by choline, and site-specific changes in DNA methylation in the CpG island of the *perk* promoter region were measured. As shown in Figure 6D, among these specific methylation sites, the −*211* site was the most sensitive to dietary choline, suggesting this site was the potential regulatory site for choline-regulating *perk* methylation.

To further confirm the above regulatory mechanism and its role on CMs assembly, IECs were incubated with the specific inhibitor of DNA methylation 5-azacytidine (AZA) in vitro. First, AZA significantly inhibited the whole DNA methylation level of *perk*, and then increased the promoter activity of *perk*, which led to the up-regulation of transcript and protein expression levels of *perk* in all groups (Figure 6E–H). This confirmed and indicated the regulatory role of DNA methylation on *perk* expression. Second, under FA/choline incubation, site-specific DNA methylation analysis indicated that the −*211* site showed the most significant changes after AZA incubation among these methylation sites in the CpG island of the *perk* promoter region (Figure 6I and Appendix A). Moreover, further site-specific mutation analysis of the *perk* promoter showed only that the mutation of −*211* methylation site could markedly increase the luciferase activity of the *perk* promoter (Figure 6J and Appendix A). Importantly, the difference of the luciferase activity between FA and choline groups was reduced after the mutation of the −*211* methylation site (Figure 6J), suggesting the involvement of −*211* methylation site in FA/choline-regulating *perk* expression. Lastly, as shown in Figure 6K–L, the mutation of −*211* methylation site caused the whole down-regulated levels of CMs-lipoprotein and TG content in all groups, and also decreased the difference between FA and choline groups. This suggested that the −*211* methylation site participated in *perk*-mediated CMs assembly under FA/choline treatment. Collectively, these results emphasized the contribution of choline acting as a methyl donor in regulating *perk* expression via controlling −*211* site-specific DNA methylation and its essential role for CMs assembly.

## 4. Discussion

The frequent occurrence of intestinal lipid dysfunction and the central role of the ER in CMs assembly prompted us to examine the mechanism underlying UPRer regulating HFD-impaired CMs assembly and its role in intestinal lipid homeostasis. The key findings from this study demonstrated that *perk*-mediated UPRer controlled HFD-caused impairment of CMs assembly by Perk-Apo48 interaction. Meanwhile, we present findings implicating the regulatory role of choline-mediated DNA methylation in CMs assembly via specific CpG sites in the promoter region of *perk*. Our study highlights choline-mediated CpG methylation of the *perk* promoter contributes to the alleviation of HFD-induced impairment of CMs assembly and intestinal lipid dysfunction.

The present study Indicated that HFD caused intestinal structural damage, as evidenced by the incremental villi height and muscular layer thickness, which is in agreement with other studies [1,45]. Then, not surprisingly, our present study also found that HFD caused excessive TG accumulation and intestinal lipid dysregulation, which was also confirmed by our previous studies [19]. We further showed that HFD-aggregated TG mainly originated from FA re-esterification, not de novo FA synthesis. This suggested, under the state of sufficient intake of dietary FA including HFD condition, enterocytes give preference to FA re-esterification for TG synthesis, similar to what other studies found [1,46]. Here, given that FA re-esterification was the main method of CMs-TG production, we next explored the effect of HFD on CMs assembly. The present study demonstrated that HFD caused impairment of CMs assembly via down-regulating CMs size, CMs-TG/apolipoproteins, and also MTP at the levels of mRNA transcription and protein translation/activity in vivo, which was somewhat similar with the findings of other studies [7,8]. In the light of CMs working as the main transfer vehicle for intestinal TG, these results suggested that HFD-induced impairment of CMs assembly was also the critical predisposition to intestinal lipid accumulation, which in turn intensified the impairment of CMs assembly. Additionally, the whole reduced level of intestinal lipid droplet accumulation caused by si-*perk*-improved CMs assembly in vitro, also supporting the above conclusion.

CMs assembly begins in the rough ER with the synthesis and initial lipidation of apolipoprotein B (apoB) [4,9,10,11]. Thus, CMs assembly depends on the normal function of the ER [4]. UPRer as a conserved mechanism in response to ER dysfunction is frequently induced by HFD [14,17]. We therefore next sought to test the role of UPRer in HFD-induced impairment of CMs assembly. The present study indicated that HFD caused massive ER swelling and up-regulated the mRNA and protein expression of UPRer markers, suggesting the occurrence of HFD-induced UPRer, similar to other studies [17]. UPRer is mediated by three pathways, the *perk-eif2α* pathway, the *ire1α-xbp1* pathway, and the *atf6* pathway [14]. Some studies, including our previous study, have indicated the differential effects and mechanisms of three UPRer branches on regulating cell function and metabolism [17,47]. The current study indicated, comparing to other two UPRer branches, the *perk-eif2α* pathway was most sensitive to HFD. This implied the *perk-eif2α* pathway played the main role in HFD-induced UPRer, which prompted us to next explore the role and mechanism of the *perk-eif2α* pathway in HFD-impairing CMs assembly.

Our current study found that *perk* knockdown promotes CMs assembly under FA incubation, which issupported by the increasing CMs’ size and CMs-TG/apolipoproteins, suggesting the upstream regulatory role of *perk* in CMs assembly. Our results also suggested that *perk*-mediated UPRer controlled FA-inhibited CMs assembly by regulating two key proteins in CMs assembly (Apob48 and Mtp) [9,11]. Similarly, Iqbal et al. [13] pointed out IRE1β, another UPRer protein, plays a role in regulating CMs production by selectively degrading *mtp* mRNA. Importantly, we next found there was a protein interaction between Perk and Apob48, which provides evidence for direct crosstalk between UPRer and CMs assembly. Interestingly, this interaction was significantly enhanced by HFD, indicating the involvement of the Perk–Apob48 interaction in HFD impairing CMs assembly. Similarly, Qiu et al. [25] found that UPRer could attenuate apolipoprotein synthesis via *perk* signaling. From these data, we suggested HFD-regulated *perk* signaling caused UPRer, which resulted in the impairment of CMs production via the Perk–Apob48 interaction.

One of the objectives of the current study was to explore the contribution of dietary choline acting as a methyl donor in alleviating HFD-induced intestinal lipid accumulation. First, as expected, we found that dietary choline alleviated HFD/FA-induced intestinal lipid metabolic disorders by improving *perk*-mediated CMs assembly, which is in agreement with other studies [33,48]. Similarly, the essential role of CMs secretion in choline alleviating intestinal lipid metabolism also been found by da Silva et al. [33]. Second, the present study pointed out dietary choline significantly up-regulated the HFD-reduced DNA methylation level of *perk* in vivo, which is consistent with other researches [49,50]. Lastly and also importantly, our in vitro study further indicated choline-increased DNA methylation level of the *perk* main via activating −*211* CpG methylation site, which in turn down-regulated the promoter transcription activity and protein expression of *perk*, and then brought about amelioration for FA-induced impaired CMs assembly. These results suggested the involvement of choline serving as a methyl donor in intestinal lipid metabolism, in agreement with other studies [51]. Similarly, in rodents, dietary restriction of the methyl donors (methionine and/or choline) rapidly and reliably induces a spectrum of liver injury histologically similar to NASH [52]. Furthermore, Sivanesan et al. [48] suggested choline improved lipid homeostasis in obesity by its participation in demethylation. Thus, our present study emphasized the critical contribution of choline acting as a methyl donor in alleviating HFD-induced intestinal lipid accumulation via CMs assembly modulated by the site-specific DNA methylation of *perk*. In addition, the close correlation between choline deficiency and NASH had been indicated in several studies [21,35,52]. Moreover, some studies suggest that blocking chylomicron assembly increases the compensatory transfer of intestinal permeability and liver lipogenesis, which in turn affects the creation of NASH [35,52]. Thus, these results provided a new insight into choline-deficient diet-induced NASH from the view of CMs assembly and choline acting as a methyldonor.

In conclusion, the present study clearly demonstrated that HFD induced UPRer main by *perk* signaling, which led to the impairment of CMs assembly though the interaction of Perk with Apob48, eventually resulting in intestinal lipid accumulation. Additionally, choline showed alleviating effects on the above-mentioned process via mediating the site-specific DNA methylation of *perk*. The detail mechanism is shown in Figure 7.

## Figures and Tables

**Figure 1 cells-11-03848-f001:**
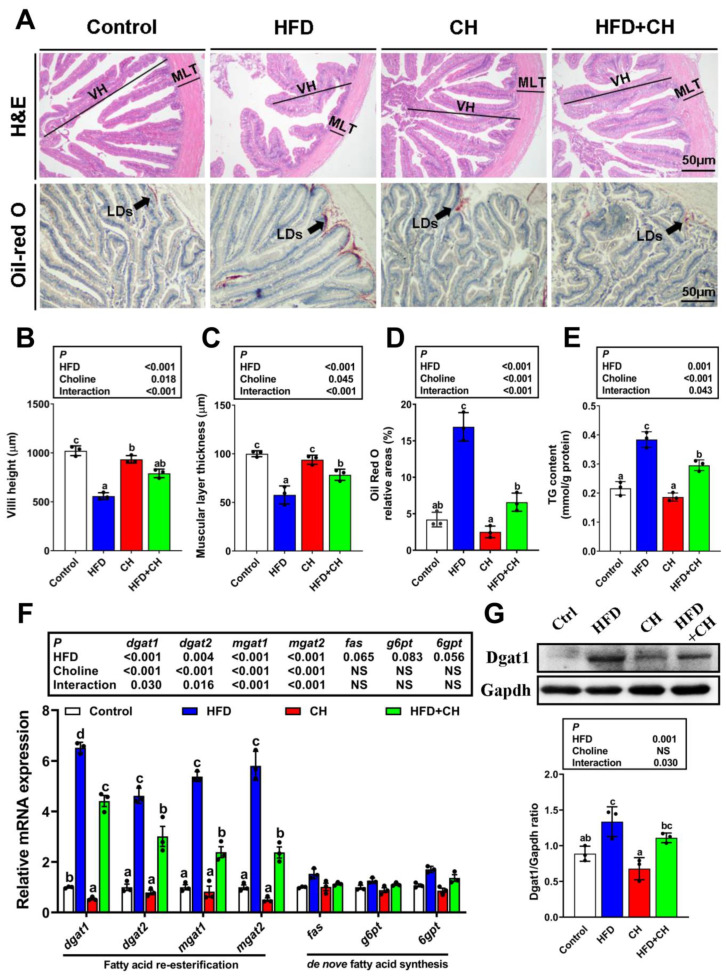
**Dietary choline alleviated HFD-induced intestinal lipid dysregulation.** (**A**) Representative images of intestinal H&E and Oil red O stained. Scale bar, 50 μm; VM, villi height. MLT, muscular layer thickness. LDs, lipid droplets. (**B**,**C**) Relative intestinal villi height and muscular layer thickness in H&E staining. (**D**) Relative areas for LDs in Oil red O staining. (**E**) Intestinal TG content. (**F**) mRNA levels of the genes related to FA re-esterification and de novo FA synthesis. (**G**) Western blot analysis and quantification analysis for Dgat1. Data are mean ± SEM, n = 3. Labeled means without a common letter differ, *p* < 0.05 (2-factor ANOVA, Duncan’s post hoc test). NS, not significant (*p* ≥ 0.05).

**Figure 2 cells-11-03848-f002:**
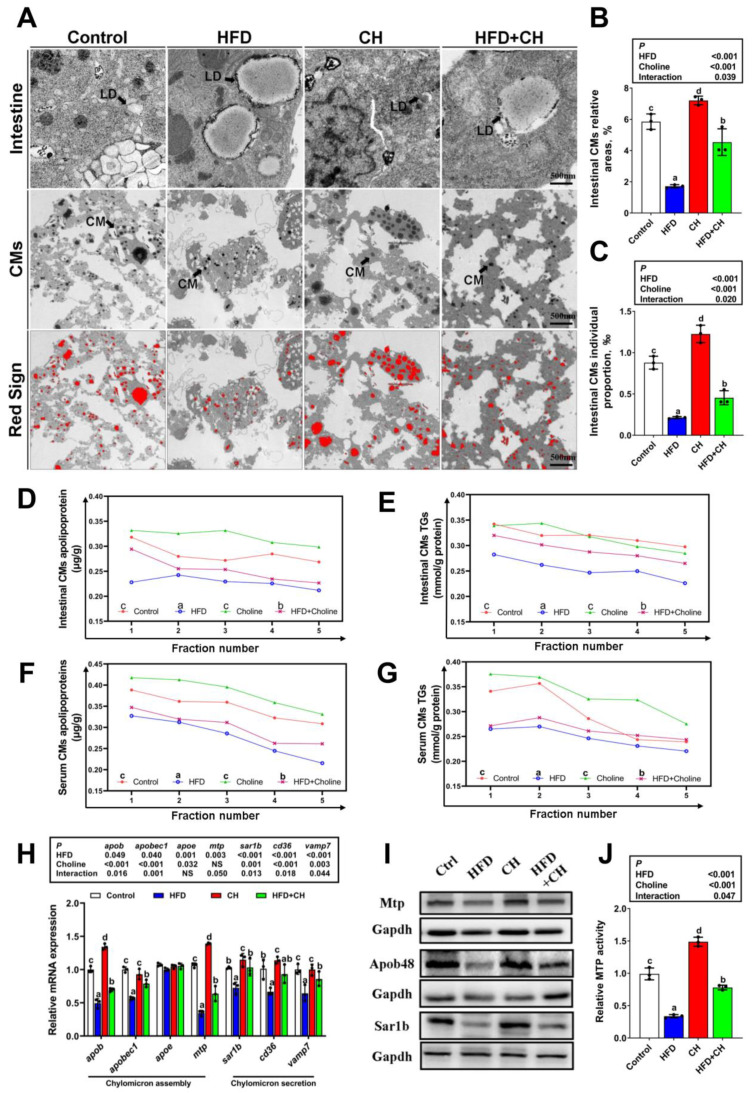
**Dietary choline improved HFD-induced impairment of CMs assembly.** (**A**) TEM structures of intestines and the isolated fractions of CMs. Scale bar, 500 nm; LD, lipid droplet. (**B**,**C**) Relative intestinal CMs areas and individual proportion in TEM structures. (**D**–**G**) Density gradient curve and quantification analysis for CMs-TGs/or apolipoprotein in intestines and serum. (**H**) mRNA levels of the genes related to CMs assembly and secretion. (**I**) Western blot for Mtp, Apob48, and Sar1b. (**J**) Relative MTP activity. Data are mean ± SEM, n = 3. Labeled means without a common letter differ, *p* < 0.05 (2-factor ANOVA, Duncan’s post hoc test). NS, not significant (*p* ≥ 0.05).

**Figure 3 cells-11-03848-f003:**
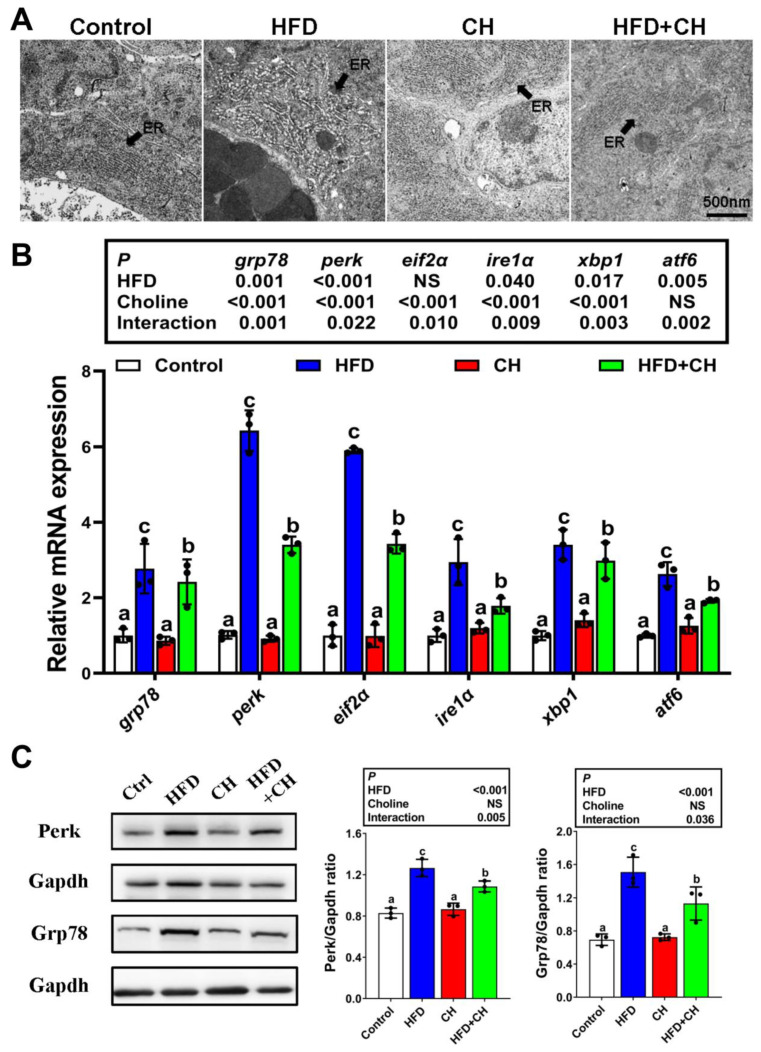
**Alleviated effects of dietary choline on HFD-induced UPRer.** (**A**) TEM structures of the intestinal endoplasmic reticulum (ER). Scale bars, 500 nm. (**B**) mRNA levels of the genes related to UPRer. (**C**) Western blot and quantification analysis for Perk and Grp78. Data are mean ± SEM, n = 3. Labeled means without a common letter differ, *p* < 0.05 (2-factor ANOVA, Duncan’s post hoc test). NS, not significant (*p* ≥ 0.05).

**Figure 4 cells-11-03848-f004:**
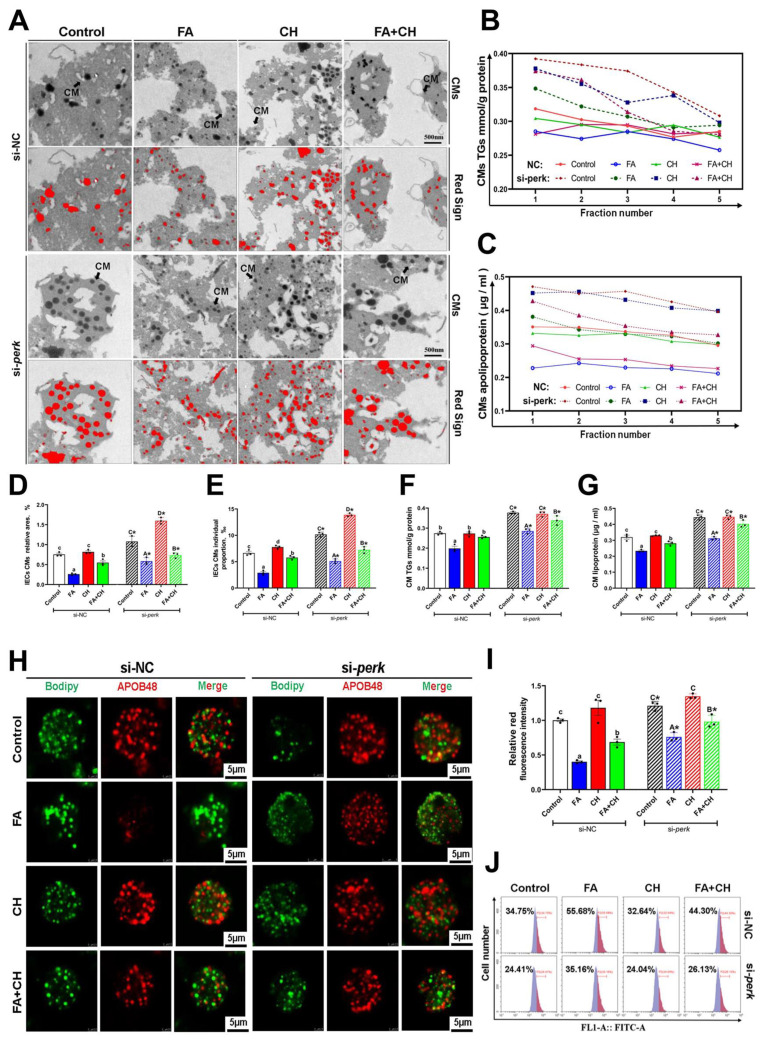
**Perk signaling regulated FA/choline-mediated CMs assembly.** (**A**) TEM structures of the isolated fractions of IECs’ CMs under FA/choline incubation with si-*perk* transfection. Scale bar, 500 nm. (**B**,**C**) Density gradient curve and quantification analysis for CMs-TGs/or apolipoprotein. (**D**,**E**) Relative CMs areas and individual proportion in TEM structures. (**F**,**G**) Relative quantification analysis for CMs-TGs/or apolipoprotein. (**H**) The co-localization of Apob48 (red) and LDs (Bodipy 493/503, green) in IECs. Scale bars, 5 μm. (**I**) Quantitative analysis for relative red intensity of fluorescence in H. (**J**) The presence of LDs with Bodipy 493/503 staining were demonstrated by flow cytometry. Data are mean ± SEM, n = 3 independent biological experiments; different lower-case letters indicate significant differences in si-NC groups; different capital letters indicate significant differences in si-*perk* groups (*p* ≤ 0.05); asterisks indicate significant differences between si-NC and si-*perk* groups (* *p* ≤ 0.05).

**Figure 5 cells-11-03848-f005:**
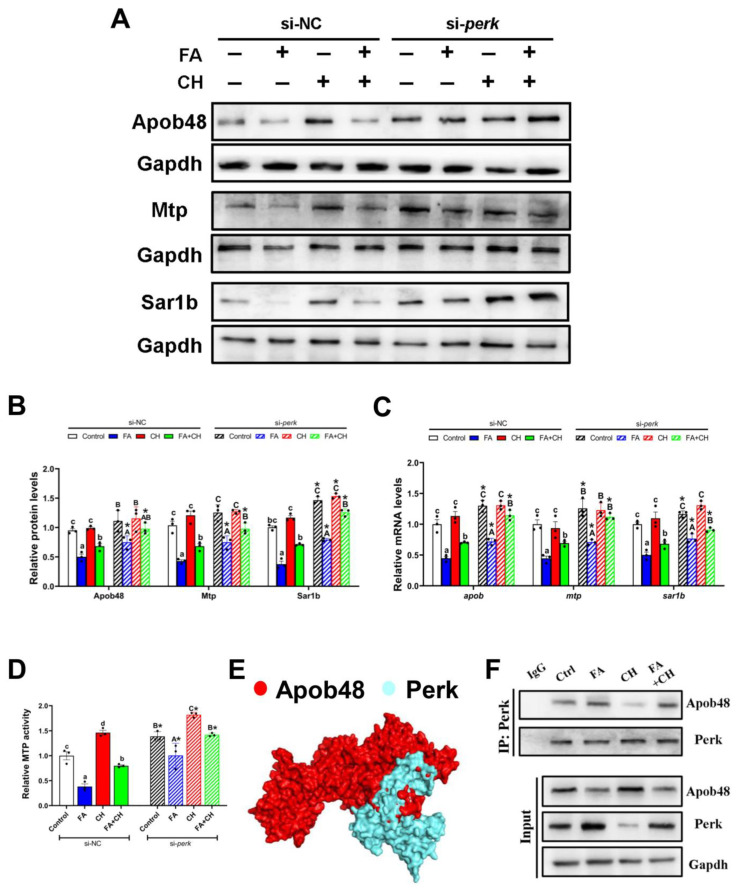
**Perk-Apob48 interaction is required for FA/choline-mediated CMs assembly.** (**A**) Western blot analysis for Mtp, Apob48, and Sar1b in the IECs under FA and CH incubation and transfected with *perk* siRNA. (**B**) Quantification analysis for Western blot. (**C**) mRNA levels of *apob, mtp, sar1b* genes. (**D**) Relative MTP activity. (**E**) The structural protein prediction model between Perk and Apob48. (**F**) IP analysis of Perk-Apob48 complex. Data are mean ± SEM, n = 3 independent biological experiments; different lower-case letters indicate significant differences in si-NC groups; different capital letters indicate significant differences in si-*perk* groups (*p* ≤ 0.05); asterisks indicate significant differences between si-NC and si-*perk* groups (* *p* ≤ 0.05).

**Figure 6 cells-11-03848-f006:**
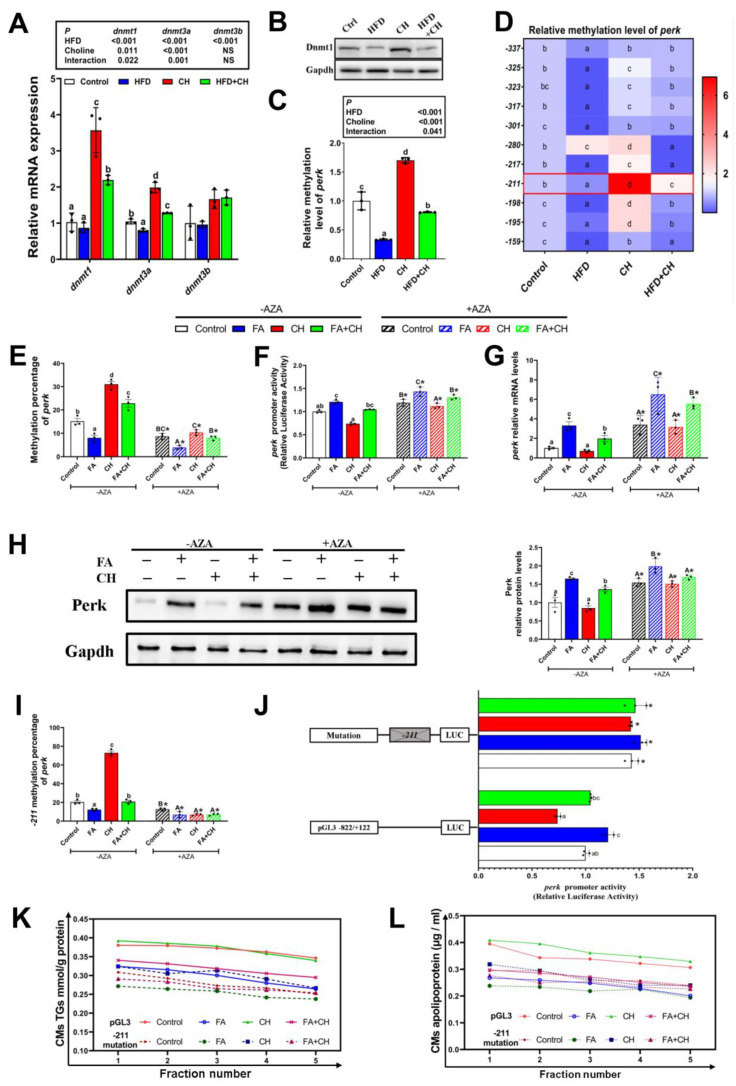
**Choline down-regulated perk expression by controlling** −***211* site-specific DNA methylation.** (**A**) mRNA levels of the genes related to DNA methyltransferase in the intestine. (**B**) Western blot for Dnmt1. (**C**) The histogram of relative methylation level of *perk*. (**D**) Heat map of relative methylation level of *perk*. (**E**) The histogram of quantitative methylation percentage of *perk* in the IECs under FA and CH incubation and transfected with 5-azacitidine (AZA). (**F**) The relative luciferase activities of *perk.* (**G**) mRNA levels of *perk.* (**H**) Western blot and quantification analysis for Perk. (**I**) The quantitative methylation percentage of *perk* −*211* methylation site. (**J**) Site mutation analysis of *perk* −*211* methylation site on pGl3-*perk* −*822/+122* vectors. (**K**,**L**) Density gradient curve and quantification analysis for CMs-TGs/or apolipoprotein in the IECs under FA and CH incubation and transfected with −*211* methylation site mutation. Data are mean ± SEM, n = 3 independent biological experiments; different lower-case letters indicate significant differences in −AZA groups; different capital letters indicate significant differences in +AZA groups (*p* ≤ 0.05); asterisks indicate significant differences between −AZA and +AZA groups (* *p* ≤ 0.05).

**Figure 7 cells-11-03848-f007:**
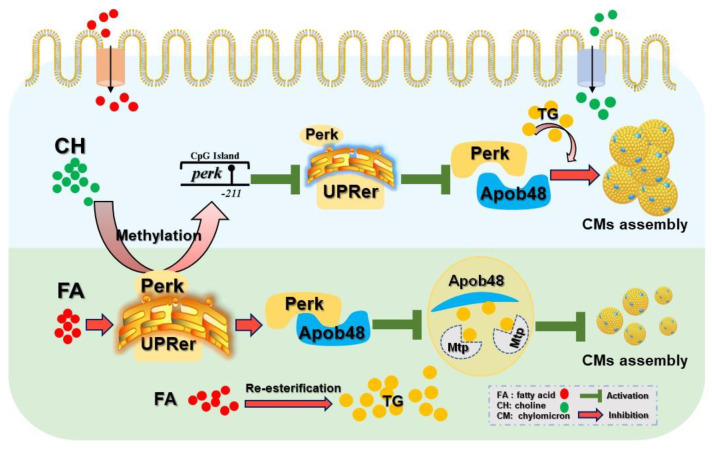
Graphical conclusions for the mechanism of dietary choline mitigates high-fat diet-impaired chylomicrons assembly via UPRer modulated by *perk* DNA methylation.

## Data Availability

Not applicable.

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
