# Peer review of "Dietary Choline Mitigates High-Fat Diet-Impaired Chylomicrons Assembly via UPRer Modulated by *perk* DNA Methylation"

_cells, 2022, doi:10.3390/cells11233848_

Round 1
Reviewer 1 Report
This paper with an attractive title for readers who concerns about effects of different diets on the lipid metabolism. Though it was a study focus on yellow catfish, the authors clarified an important regulatory role of Perk, in the HFD impairing CMs assembly by protein-protein interaction with Apo48. Furthermore, they found a novel alleviating function of choline in this process via site-specific DNA methylation of perk, which provides a better understanding of the molecular mechanism of CMs assembly and secretion in HFD-induced lipid metabolic dysfunction. However, there are still some concerns:
1. The effects of choline on mammals were reported in multiple papers, and the choline deficient diet was widely used for mice models in the studies of NAFLD or NASH. Did the conclusion of this paper shed new lights for lipid metabolism in NAFLD?
2. The manuscript needs to be carefully reviewed before submission. The statistics indication in figures made readers confused. Please replaced with a more straightforward description.
3. The amount of “GAPDH” expression varies a lot in several figures? Would the different treatment affect its’ expression?
4. The origin images of western blot should contain all the blots used for quantification.
5. Figure 7 is missing.
Author Response
Reviewer #1
This paper with an attractive title for readers who concerns about effects of different diets on the lipid metabolism. Though it was a study focus on yellow catfish, the authors clarified an important regulatory role of Perk, in the HFD impairing CMs assembly by protein-protein interaction with Apo48. Furthermore, they found a novel alleviating function of choline in this process via site-specific DNA methylation of perk, which provides a better understanding of the molecular mechanism of CMs assembly and secretion in HFD-induced lipid metabolic dysfunction. However, there are still some concerns:
Comment 1: The effects of choline on mammals were reported in multiple papers, and the choline deficient diet was widely used for mice models in the studies of NAFLD or NASH. Did the conclusion of this paper shed new lights for lipid metabolism in NAFLD?
Response 1: Thank you very much for your specific comments. We totally agree with you here. There is closely relationship between choline-deficient diet and NAFLD. So based on your comments, we added more information about this and make new conclusion for this paper. For the detail, please see the text.
Comment 2: The manuscript needs to be carefully reviewed before submission. The statistics indication in figures made readers confused. Please replaced with a more straightforward description.
Response 2: Thank you very much for your suggestion. Based on your important comments, we have improved the statistics indication. In vitro study, we used different lower-case letter to indicate significant differences in si-NC / -AZA groups, and used different capital letter to indicate significant differences in si-perk / +AZA groups. We hope this way will provide more straightforward description. For the detail, please see the text. If you have any other comments for here, please do not hesitate to contact us. Thanks again!
Comment 3: The amount of “GAPDH” expression varies a lot in several figures? Would the different treatment affect its’ expression?
Response 3: Thank you very much for your specific comments. Here, we also think the “GAPDH” expression are not so stable if from our previous pictures in the article, especially for Figure 5. But based on all the result from our origin images of western blot, the “GAPDH” expression didn’t varies same in all repetitions, just some repetition was not stable. This may due to our operating error. Here, we checked all “GAPDH” expression in our article and replaced more stable “GAPDH” expression in Figure 5. Meanwhile, we have provided all these origin images of western blot in our supplementary materials. For the detail, please see the text. We hope that you are satisfactory to our revisions. But if you have any other comments about this, we can repeat more of our WB experiments and provided more stable “GAPDH” expression in Figures. Thanks again!
Comment 4: The origin images of western blot should contain all the blots used for quantification.
Response 4: Thank you very much for your important comments. Based on your important comment, we have added all the blots used for quantification in origin images. For the detail, please see the text.
Comment 5: Figure 7 is missing.
Response 5: Sorry for losing these materials. We have submitted Figure 7 in system again and hope you can see them. For the detail, please see the text.
Reviewer 2 Report
This study investigated choline as a methyl donor to inhibit perk expression, thereby alleviating HFD-induced ER UPRer and regulating CMs assembly, and also demonstrated the direct interaction between Perk and Apob48 during this process, ultimately reducing lipid accumulation in the intestine. The introduction provide sufficient background and the methods are adequately described.
However, the figures provided in the ms are unsatisfactory.
1. All the original-images of WB were not provided with the whole membrane. And the original-images of Fig 4 first Gpadh may be not correct.
2. Where were the supplementary materials including Tables and Figures? How about the Fig 7?
3. The Fig 2D and 2E were same?
4. The structure of ER was not clear in image of HFD+CH in Fig 3A.
5. How to explain the same trendlines of Fig 4B and Fig 6K? There were the different experiments.
And a minor problem is that in line 110-112, different quality of choline chloride added to control and HFD group, 563.4 and 578.9 mg per kg diet, respectively. Why didn’t add as same?
Author Response
Reviewer #2
This study investigated choline as a methyl donor to inhibit perk expression, thereby alleviating HFD-induced ER UPRer and regulating CMs assembly, and also demonstrated the direct interaction between Perk and Apob48 during this process, ultimately reducing lipid accumulation in the intestine. The introduction provides sufficient background and the methods are adequately described.
However, the figures provided in the ms are unsatisfactory.
Comment 1: All the original-images of WB were not provided with the whole membrane. And the original-images of Fig 4 first Gpadh may be not correct.
Response 1: Thank you for pointing the mistake. Based on your important comments, we have provided all these original-images for statistical analysis. As you mention first Gpadh in Fig 4, which we think here maybe you mean the Fig.5, have been corrected and also the original-images were provided. For the detail, please see the text.
Comment 2: Where were the supplementary materials including Tables and Figures? How about the Fig 7?
Response 2: Sorry for losing these materials. We have submitted these in system again and hope you can see them. For the detail, please see the text.
Comment 3: The Fig 2D and 2E were same?
Response 3: Fig 2E should be the density gradient curve and quantification analysis for CMs-TGs in intestine. We have revised them. For the detail, please see the text.
Comment 4: The structure of ER was not clear in image of HFD+CH in Fig 3A.
Response 4: Based on your important comment, we have provided another more clear image for HFD+CH in Fig 3A. For the detail, please see the text.
Comment 5: How to explain the same trendlines of Fig 4B and Fig 6K? There were the different experiments.
Response 5: Thank you very much for your constructive comment. Fig 4B and Fig 6K should have different as you mentioned. But here we used the wrong image for Fig 6K and we have revised it with right one, which also have the similar trends. For the detail, please see the text.
Comment 6: And a minor problem is that in line 110-112, different quality of choline chloride added to control and HFD group, 563.4 and 578.9 mg per kg diet, respectively. Why didn’t add as same?
Response 6: Thank you for your suggestion. For these two groups, choline chloride was added at the same levels. However, here the concentration of 563.4 and 578.9 mg per kg diet were based on the actual measured results, which will be some measurement errors. Also this same situation was been found in our previous study (Song et al., 2022).
References:
Song YF, Zheng H, Luo Z, Hogstrand C, Bai ZY, Wei XL. Dietary Choline Alleviates High-Fat Diet-Induced Hepatic Lipid Dysregulation via UPRmt Modulated by SIRT3-Mediated mtHSP70 Deacetylation. Int J Mol Sci. 2022 Apr 11;23(8):4204. doi: 10.3390/ijms23084204.
Reviewer 3 Report
Congratulations to the authors for this manuscript that contains a very good work, with complicated techniques such as western, histology, RTqPCR, cell cultured, small interference RNA transfection and lipoproteins isolation among others. The figures have a lot of quality and the message is very clear, which is why it should be accepted in Nutrients.
Author Response
Reviewer #3
Comment: Congratulations to the authors for this manuscript that contains a very good work, with complicated techniques such as western, histology, RTqPCR, cell cultured, small interference RNA transfection and lipoproteins isolation among others. The figures have a lot of quality and the message is very clear, which is why it should be accepted in Nutrients.
Response: Thank you for your encouragement and positive comments, which have given us confidence in future research.
Reviewer 4 Report
cells-2040614-peer-review-v1
In my opinion this paper is interesting, innovative and providing valuable information. The discussed subject is not my principal area of research, and I will appreciate if my opinion will be considered as secondary, and giving priority of the other reviewers. However, description of the introduction, material and methods and following results and discussions were well described. Paper was easy to follow.
I would like to suggest that present paper can be accepted for publication, however, some minor corrections will need to be taken into accounts:
Please, in the entire manuscript use same style for presentation. Choice mg/L or mg L-1, but do not mix them.
Maybe more appropriate titles for 2.2. and 2.3. can be suggested.
Ln170. Maybe will be more appropriate if can be written: "in our previous publications". To avoid misunderstanding that is refer to present publication and use plural, since is referring to 2 papers.
Please, for material and equipment used in this study, provide name of the company and address, included: city, state in abbreviate form (in case of federal countries) and country name. Please, in second occasion, only name of the company needs to be provided. Use address of headquarter, and not of local distributors.
Ln187: Add additional information - City, country.
Ln190: add USA after NY.
Lb195: Provide information for suppler of the kit.
Ln235: Please, provide conditions of centrifugation.
Ln427: do not need italics for Fig.
Figure 7 was not provided.
Author Response
Reviewer #4
In my opinion this paper is interesting, innovative and providing valuable information. The discussed subject is not my principal area of research, and I will appreciate if my opinion will be considered as secondary, and giving priority of the other reviewers. However, description of the introduction, material and methods and following results and discussions were well described. Paper was easy to follow.
I would like to suggest that present paper can be accepted for publication, however, some minor corrections will need to be taken into accounts:
Comment 1: Please, in the entire manuscript use same style for presentation. Choice mg/L or mg L-1, but do not mix them.
Response 1: Thanks for your valuable comments. We have unified same style in all units. For the detail, please see the text.
Comment 2: Maybe more appropriate titles for 2.2. and 2.3. can be suggested.
Response 2: Thanks for your important comments. We have changed the titles of 2.2. and 2.3. For the detail, please see the text.
Comment 3: Ln170. Maybe will be more appropriate if can be written: "in our previous publications". To avoid misunderstanding that is refer to present publication and use plural, since is referring to 2 papers.
Response 3: Based on your important comments, we have modified these descriptions in Materials and methods. For the detail, please see the text.
Comment 4: Please, for material and equipment used in this study, provide name of the company and address, included: city, state in abbreviate form (in case of federal countries) and country name.
Response 4: Thank you very much for your constructive comment. We have provided these informations in Materials and methods. For the detail, please see the text.
Comment 5: Please, in second occasion, only name of the company needs to be provided. Use address of headquarter, and not of local distributors.
Response 5: Thank you very much for your important comments and advice. We have provided these informations in Materials and methods. For the detail, please see the text.
Comment 6: Ln187: Add additional information - City, country.
Response 6: Thank you very much for your important comments. We have added this information in Ln187. For the detail, please see the text.
Comment 7: Ln190: add USA after NY.
Response 7: Thank you very much for your exact comments. We have added this information in Ln187. For the detail, please see the text.
Comment 8: Lb195: Provide information for suppler of the kit.
Response 8: We have added these informations. For the detail, please see the text.
Comment 9: Ln235: Please, provide conditions of centrifugation.
Response 9: Thank you very much for your important comments. We have added the conditions of centrifugation in Ln235. For the detail, please see the text.
Comment 10: Ln427: do not need italics for Fig.
Response 10: Thanks for your important comments. We have revised them in Ln427. For the detail, please see the text.
Comment 10: Figure 7 was not provided.
Response 10: Thank you very much for your constructive comments. We have added the Figure 7. For the detail, please see the text.
Round 2
Reviewer 1 Report
The authors have responded to my questions. The manuscript is improved after revised. There is no more comment from me.
Author Response
Thank you for your encouragement and positive comments, which have given us confidence in future research. We are also glad that you are satisfied with our reply. Thanks again!
Reviewer 2 Report
Some minor points.
1. In line 234 and 242, the units should be “g”, not “g min-1”.
2. Table S2 was not mentioned in ms.
The original images should be carefully checked and organized, especially Fig4 in the file “cells-2040614-original-images” (maybe it should be Fig5).
Author Response
Reviewer #2
Comment 1: In line 234 and 242, the units should be “g”, not “g min-1”.
Response 1: Thank you very much for your constructive comments. We have revised this error in Ln234 and 242. For the detail, please see the text.
Comment 2: Table S2 was not mentioned in ms.
Response 2: Thanks for your valuable comments. We have added these informations in Materials and Methods section. For the detail, please see the text.
Comment 3: The original images should be carefully checked and organized, especially Fig4 in the file “cells-2040614-original-images” (maybe it should be Fig5).
Response 3: Thank you very much for your important comments and advice. Based on your important comments, we have checked and corrected the original-images, and the mistake of Fig 5 has been revised. For the detail, please see the text.